# Reshaping the Pancreatic Cancer Microenvironment at Different Stages with Chemotherapy

**DOI:** 10.3390/cancers15092448

**Published:** 2023-04-25

**Authors:** Maozhen Peng, Ying Ying, Zheng Zhang, Liang Liu, Wenquan Wang

**Affiliations:** 1Department of Pancreatic Surgery, Zhongshan Hospital, Fudan University, Shanghai 200032, China; pengmz2022@163.com (M.P.); 18767682351@163.com (Y.Y.); zhengzhang21@m.fudan.edu.cn (Z.Z.); 2Cancer Center, Zhongshan Hospital, Fudan University, Shanghai 200032, China; 3Department of General Surgery, Zhongshan Hospital, Fudan University, Shanghai 200032, China

**Keywords:** pancreatic cancer, chemotherapy, neoadjuvant chemotherapy, tumor microenvironment

## Abstract

**Simple Summary:**

Reshaped pancreatic tumor microenvironment by chemotherapy, ultimately preventing or promoting tumor progression, is presented by the alteration of malignant, stromal cells, and immune cells, in a quantitative, functional, and spatial manner. They were studied predominantly in two cohorts, which include pancreatic cancer with borderline resectable and locally advanced disease (and minor resectable disease) under neoadjuvant chemotherapeutic regimens and most resectable and metastatic disease under adjuvant chemotherapeutic regimens.

**Abstract:**

The dynamic tumor microenvironment, especially the immune microenvironment, during the natural progression and/or chemotherapy treatment is a critical frontier in understanding the effects of chemotherapy on pancreatic cancer. Non-stratified pancreatic cancer patients always receive chemotherapeutic strategies, including neoadjuvant chemotherapy and adjuvant chemotherapy, predominantly according to their physical conditions and different disease stages. An increasing number of studies demonstrate that the pancreatic cancer tumor microenvironment could be reshaped by chemotherapy, an outcome caused by immunogenic cell death, selection and/or education of preponderant tumor clones, adaptive gene mutations, and induction of cytokines/chemokines. These outcomes could in turn impact the efficacy of chemotherapy, making it range from synergetic to resistant and even tumor-promoting. Under chemotherapeutic impact, the metastatic micro-structures in the primary tumor may be built to leak tumor cells into the lymph or blood vasculature, and micro-metastatic/recurrent niches rich in immunosuppressive cells may be recruited by cytokines and chemokines, which provide housing conditions for these circling tumor cells. An in-depth understanding of how chemotherapy reshapes the tumor microenvironment may lead to new therapeutic strategies to block its adverse tumor-promoting effects and prolong survival. In this review, reshaped pancreatic cancer tumor microenvironments due to chemotherapy were reflected mainly in immune cells, pancreatic cancer cells, and cancer-associated fibroblast cells, quantitatively, functionally, and spatially. Additionally, small molecule kinases and immune checkpoints participating in this remodeling process caused by chemotherapy are suggested to be blocked reasonably to synergize with chemotherapy.

## 1. Introduction

Pancreatic cancer is a highly fatal disease with a low 5-year survival rate, making it an increasingly common cause of cancer mortality. Clinical staging classifies patients with pancreatic cancer into resectable, borderline resectable, locally advanced, and metastatic disease. All of which can be firstly treated with either neoadjuvant chemotherapy or adjuvant chemotherapy. FOLFIRINOX (comprising oxaliplatin, irinotecan, fluorouracil, and leucovorin) and AG (gemcitabine/nab-paclitaxel) regimens are recommended as neoadjuvant chemotherapeutic regimens for borderline resectable and locally advanced disease due to increased R0 (margin-negative resection) incidence and improved survival [1,2,3], as well as adjuvant chemotherapeutic regimens for advanced/metastatic disease due to higher response rates and longer median overall survival compared with gemcitabine monotherapy [4,5], respectively. Notably, consensus about the use of neoadjuvant chemotherapy on resectable disease has not been achieved due to its contrary outcomes. Other benefits of chemotherapy also include the expansion of the resectable population, reduction of tumor burden, control of occult metastases and prevention of recurrence/metastasis. (Neoadjuvant) FOLFIRINOX was associated with a 4.9- and 2.0-month improvement in survival compared with (neoadjuvant) AG for resectable/borderline resectable pancreatic cancer and metastatic pancreatic ductal adenocarcinoma, respectively [6,7]. These chemotherapeutic drugs exert anticancer effects by damaging DNA and RNA and inhibiting thymidylate synthase. Cellular uptake of 5-FU and gemcitabine could be mediated by the organic anion transporter 2 and human equilibrative nucleoside transporter 1, respectively [8,9]. High transporter expression correlates with good response to chemotherapy, making them predictive markers to guide treatment decisions.

The tumor microenvironment (TME) of pancreatic cancer, mainly comprising cancer cells, cancer-associated fibroblasts (CAFs), and immune cells, is dynamic during natural tumor progression and chemotherapeutic treatment (Figure 1) [10,11,12]. The number of ductal cells gradually decreased with tumor stage (I–II–III) progression (44.88%–36.04%–18.95%) [11]. Ductal cells in early pancreatic cancer mainly exhibit an epithelial expression profile, whereas ductal cells in later pancreatic cancer are featured by mesenchymal markers and have higher expression levels of cancer stem cell (CSC)-related genes [11]. The amount of CAFs consisted of fibroblasts, and stellate cells decreased gradually with tumor stage (I–II–III) progression (23.49%–17.09%–15.12%) [11]. Complement-secreting CAFs (csCAFs) are located in the tissue stroma adjacent to malignant ductal cells in only stage I/II pancreatic cancer [11], while pancreatic stellate cells (PSCs) only locate in the microenvironment of stage III pancreatic cancer. Trajectories based on transcriptional similarities suggested the evolution from csCAFs towards PSCs. The immune microenvironment is featured by a prominent and continuous presence of effector lymphocytes in the precursor stages (acinar-to-ductal metaplasia and pancreatic intraepithelial neoplasia). Consistently, Bernard et al. [13] demonstrated a high proportion of cytotoxic T cells were observed in low-grade IPMNs compared to high-grade IPMNs and invasive pancreatic cancer. During invasive pancreatic cancer progression, stage II/III pancreatic cancer presented more cytotoxic T cells, effector T cells, and memory T cells than stage I [11]. Simultaneously, more Tregs and exhausted T cells also accumulated in the later stages to suppress activated lymphocytes by downregulating IFNG expression. In addition, stage III pancreatic cancer had more M1, M2, and TAM (tumor-associated macrophages) than earlier-stage pancreatic cancer. Naïve B cells, which appeared in stage II but not stage I, gradually repolarize to those with pro-tumor or anti-tumor function in stage III [11]. Immature Ly-6C+ monocytes repolarize from a BST2+/MHC-II + phenotype to an Arg-1 + phenotype over time [12].

Progressed pancreatic cancer is attributed to the coordinated evolution of malignant and non-malignant cells [14]. Ductal cells with high proliferation gene signatures are associated with sporadic and exhausted CD8+ T cell infiltration [15]. Proliferated inhibitors targeting pancreatic cancer cells could also significantly induce CD8+ T cell infiltration. Inflammatory CAFs, which often appear in the later stage, play roles in recruiting macrophages in a CXCL12–CXCR4-dependent manner and in hindering cytotoxic T cell recruitment into tumor sites, creating the notorious immune-suppressive TME. Additionally, sub-tumor microenvironments (sub-TMEs) have been proposed to validate their roles in propelling malignant cell progression [14]. “Reactive” sub-TMEs, inhabited by aggressive tumor cell phenotypes, are featured by complex but functionally coordinated fibroblast and immune hot communities (rich in immune cell infiltration). The matrix-rich “deserted” sub-TMEs harbor fewer activated fibroblasts, while tumor-suppressive features also are markedly chemoprotective. Dynamic profiling of the tumor microenvironment suggests that the TME reshaping ability of the same chemotherapeutic regimen is inconsistent at different stages since the TME could, in turn, impact the chemotherapeutic functions. For example, later stage pancreatic cancer is dominant in tumor-associated macrophages and inflammatory CAFs, which have been demonstrated to mediate chemoresistance to protect cancer cells [16].

Taken together, the reshaped TME under chemotherapy on a given tumor stage may be more proper to reflect real alterations since this strategy excludes an impact of baselines. Notably, genetic mutation and bacteria could also contribute to the TME heterogeneity [17,18]. Distinct T-cell, natural killer, macrophage, and dendritic cell populations enrich in BRCA2-deficient but not BRCA1-deficient tumors [19]. Bacterial communities populate microniches that are less vascularized, highly immuno-suppressive, and associated with malignant cells with lower levels of Ki-67 as compared to bacteria-negative tumor regions [20]. Chemotherapeutic courses may also impact the evolution of malignant and non-malignant cells by early immunogenic cell death, selection of adaptive clones, and even induction of gene mutation [21,22,23,24,25]. Available targets during cancer development are also various. Some relatively small molecule kinase inhibitors (SMKIs) and immune checkpoint inhibitors (ICIs) are listed in the section of “Targeted therapy to synergize with chemotherapy”.

In this review, the reshaped pancreatic TME by chemotherapy, quantitatively, functionally, and spatially, is presented by the alteration of malignant, stromal cells, and immune cells in two cohorts including pancreatic cancer with borderline resectable and locally advanced disease (and minor resectable disease) under neoadjuvant chemotherapeutic regimens and most resectable and metastatic disease under adjuvant chemotherapeutic regimens. In addition, matched available targets are also mentioned for picking out reasonable regimens to synergize with chemotherapy.

The pancreatic cancer microenvironment comprises pancreatic cancer cells, stromal cells, and immune cells. These cells undergo adaptive responses due to the stress of chemotherapy. The numbers of cancer stem cells and cells with an EMT-phenotype increase to adapt to the chemotherapeutic stress, although most tumor cells fail to survival and undergo apoptosis. Additionally, increased conventional CD4 T cells and CD8 T cells are recruited to the tumor sites with closer co-locations; meanwhile, immunosuppressive cells such as regulatory T cells, tumor-associated macrophages, and MDSCs gradually decrease due to chemotherapy. More stroma and inflammatory CAFs are associated with the remaining tumor cells supporting and chemoresistance after exposure to chemotherapy.

## 2. An Anti-Tumorigenic Immune Microenvironment by Neoadjuvant Chemotherapy in Borderline Resectable and Locally Advanced Pancreatic Cancer

### 2.1. Anti-Tumor Immune Microenvironment Induced by Immunogenic Cell Death

Besides its direct apoptosis-promoting role, neoadjuvant chemotherapy with short-term induces an anti-tumorigenic microenvironment predominantly due to immunogenic cell death (Figure 2) rather than novel genomic mutations. By sequencing paired pretreatment and on-treatment gastric cancer, Kim et al. demonstrated that two cycles of platinum-based chemotherapy did not induce genomic changes but induced innate immune response and immunogenic cell death with RNA expression of genes such as ANXA1, HMGB1, CGAS, and STING [25]. Consistently, no differentially expressed genes in pancreatic cancer were identified between the treatment naïve arm and neoadjuvant FOLFIRINOX arms with 1.5–2.0 cycles [22]. However, differentially expressed genes were observed between matched pre- and post-treatment samples from gastric cancer patients receiving 5-fluorouracil + oxaliplatin-based neoadjuvant chemotherapy for 2–4 cycles [23]. These results suggest that regimen and/or course may significantly determine the evolution of the TME. Changes in immune components between AG and FOLFIRINOX regimens are not unexpectedly remarkable [26], supporting their general inflammatory response due to immunogenic cell death. These outcomes make the chemotherapeutic course an important factor that affects the TME remodeling ability of neoadjuvant chemotherapy. To date, numerous studies explore the TME remodeling ability of chemotherapy by comparing its component, function, and distribution in pre- and post-treatment arms, but no study explores the remodeling ability with different cycles.

### 2.2. Recruited Anti-Tumor Immune Cells from Progenitor Cell Differentiation

Chemotherapy could affect peripheral immunity to indirectly reshape the pancreatic TME. Gemcitabine upregulates the proportion of megakaryocyte-erythroid progenitors in the hematopoietic stem and progenitor cells by depleting other cell types and enhancing their proliferation (Figure 2). This will result in increased killing activity and infiltration of CD8 + T cells and NK cells, both in tumor and peripheral blood, partly through megakaryocyte-erythroid progenitors-secreted CCL5 and CXCL16 [27]. This result is consistent with others revealing that neoadjuvant chemotherapy with FOLFIRINOX enhances effector T cells and downregulates suppressor cells in peripheral blood. Heiduk et al. [10] found that the tendency of conventional CD4+ T cells and Tregs (the frequency of conventional CD4+ T cells was increased, and the proportion of Tregs was reduced) was similar in the TME and peripheral blood in neoadjuvant FOLFIRINOX-treated arm compared with treatment naïve arm, further suggesting that peripheral situations could partly reflect the TME conditions. It makes it possible and convenient for us to explore dynamic profiling of the TME during neoadjuvant chemotherapy courses in the same patient since blood is easier to obtain than matched tumor specimens.

An altered immune cell infiltration is derived by neoadjuvant chemotherapy in patients with borderline resectable and locally advanced disease due to chemotherapeutic immunogenic cell death and progenitor cells differentiation into anti-tumor immune cell types. Reshaped tumor microenvironment and peripheral blood is featured by increased CD8 T cells, conventional CD4 T cells, NK cells, dendritic cells, and M1-skewed tumor-associated macrophages and decreased Tregs, myeloid-derived suppressive cells, M2-skewed tumor-associated macrophages, and anti-tumor cells near to tumor cells spatially. However, inflammatory CAFs, which are associated with maintaining an immunosuppressive environment, also increased during neoadjuvant chemotherapy.

### 2.3. Reshaped Tumor Immune Cells in Quantity, Space and Function

Quantitatively, numerous studies demonstrated that neoadjuvant chemotherapy could elevate cytotoxic lymphatic cell (i.e., CD8+ T cells, conventional CD4+ T cells, and NK cells) and decline immunosuppressive myeloid cells (M2-skewed cells and myeloid-derived suppressor cells) infiltration, creating an anti-tumorigenic microenvironment [10,26,28,29] and a tendency to reverse natural tumor progression in pancreatic cancer.

Spatially, the distribution of the anticancer cells is closer to the tumor cells, while the pro-tumor cells are more distant from tumor cells in the treated arm [30]. Spatial distribution of cytotoxic CD8 T cells in proximity to cancer cells correlates with increased overall patient survival [31]. M1-polarized macrophages were located closer to tumor cells after neoadjuvant chemotherapy, and colocalization of M1-polarized macrophages and tumor cells was associated with greater tumor pathologic responses and improved patient survival [29]. Okubo et al. [32] demonstrated that CD204+ macrophage levels in the cancer stroma were similarly independent of chemotherapy, whereas those in the cancer cell nests were lower in the treated group than in the naïve group. The CD204+ macrophage count in the cancer cell nest rather than in the stroma is an independent predictor of early cancer recurrence after neoadjuvant chemotherapy. Additionally, B cells are significantly lower in immune-rich regions of tumors and unaltered in tumor and stromal cell-rich regimens after exposure to FOLFIRINOX [22].

Functionally, neoadjuvant chemotherapy induces proinflammatory cytokines and decreases pro-tumor cytokines from tumor-infiltrating T cells, with enhanced TNF-α and IL-2 and reduced IL-4 and IL-10 expression [10]. In addition, some immune subpopulations are repolarized anti-tumor ones with elevated M1/M2-skewed macrophages and conventional CD4+ T/Treg cells [29]. Immune cells in the chemotherapy arm, compared with the surgery arm, showed higher levels of the activated T-cell marker CD44 and lower levels of the immunosuppressive checkpoint molecule VISTA and TIGIT on CD8 T, Tregs, and exhausted CD4+ T cells [22,33].

With the popularization of single-cell RNA sequencing and spatial transcriptomics, increasing studies showing the TME remodeling ability of chemotherapy in aspects of quantity, space, and function will be published to support the aforementioned facts.

### 2.4. Responsive Biomarkers in Reshaped Tumor Immune Microenvironment

The reshaped TME could predict the pathologic response following chemotherapy in pancreatic cancer. Higher pathological complete response rates collate with altered distribution of overall T cells and PD-1+CD4+ T cells from the stroma to the intratumor [34]. Colocalization of M1-polarized macrophages after FOLFIRINOX (i.e., M1-polarized macrophages were located closer to tumor cells) is associated with greater tumor pathologic response and improved patient survival. The elevated neutrophil-to-lymphocyte ratio in the TME during neoadjuvant chemotherapy could impair its anti-tumor efficacy due to upregulated neutrophil-CAFs-tumor cell IL-1β/IL-6/STAT-3 signaling, showing a poor/absent response [35]. Immune characteristics in responders’ peripheral blood are similar to those in the TME under neoadjuvant chemotherapy to some extent, with increased effector T cells and decreased suppressor cells [10]. Responders to FOLFIRINOX include significantly less Treg, more total CD8 T cells, and CD27-Tbet+ effector/effector memory subsets of CD4 and CD8 T cells in peripheral blood [28]. Serum levels of IL-6 and C-reactive protein can predict the poor efficacy of mFOLFIRINOX in patients with advanced pancreatic cancer [36]. In addition, the tumor proliferation rate, together with the activation of the stroma, is also an independent prognostic factor for neoadjuvant-treated pancreatic cancer [26].

Borderline resectable and locally advanced pancreatic cancer could be transformed into resectable ones by neoadjuvant chemotherapy; meanwhile, their characteristics of the TME are also significantly changed. It is suggested that the latest characteristics should be used to guide the subsequent adjuvant treatment strategies. Most importantly, responsive biomarkers in peripheral blood are suggested to be monitored dynamically.

## 3. Immunosuppressive Myeloid Cell Recruitment and Metastasis by Chemotherapy

Adjuvant chemotherapy for pancreatic cancer patients who have undergone upfront surgery is to prevent tumor recurrence and metastasis. However, increasing studies suggest that chemotherapy could also induce cancer metastasis, which may be due to the undesirable TME response and host response to chemotherapy (Figure 3).

Chemotherapy has several effects. On the one hand, tumor cells become more adaptive and invasive and secrete CCL12 to attract CCR4^+^ TIE2^+^VEGF^+^ macrophages through the space between endothelial cells. This increases the quantity of circulating tumor cells and changes the microanatomical structure, composed of a tumor cell, a perivascular macrophage, and an endothelial cell, which is associated with tumor metastasis. On the other hand, chemotherapy creates a more immunosuppressive microenvironment for the circulating tumor cells by recruiting tumor-associated macrophages, myeloid-derived suppressor cells, and neutrophils. This is the result of increased CSF1 secreted by cancer-associated fibroblasts (CAFs), Ca3/Ca5 secreted by CAFs, and CCL2 secreted by (1) normal cells that are inhibited in micro-metastatic foci and (2) inhibited metastatic tumor cells.

### 3.1. Primary Tumor Immune Microenvironment Reshaped by Chemotherapy

The microanatomical TME of metastasis (TMEM), which comprises a tumor cell, a perivascular macrophage, and an endothelial cell, is demonstrated to be associated with pancreatic cancer metastasis [37]. Neoadjuvant chemotherapy with paclitaxel could induce the TMEM formation and upregulate its numbers [37]. Paclitaxel transforms tumor cells into invasive ones, expressing actin-regulatory protein mammalian-enabled protein and decreasing pericyte coverage, resulting in more invasive circulating tumor cells leaking into peripheral blood. Additionally, it mobilizes bone marrow-derived TIE2+ monocyte progenitors into the TMEM, in which these immature cells transform into TIE2^hi^/VEGF^hi^ macrophages [37,38]. These macrophages may be the same as the M2-skewed tumor-associated macrophages (TIE2+CXCR4^hi^VEGFA+) described by Hughes and colleagues [39]. These perivascular immunosuppressive macrophages and myeloid-derived suppressor cells could also be recruited by chemokine CXCL12 and granulocyte-macrophage colony-stimulating factor (GM-CSF) induced by pancreatic cancer cells treated with paclitaxel and gemcitabine/5-FU, respectively [39,40]. Tie2 inhibitor, CXCR4 inhibitor, and antibody against GM-CSF were demonstrated to be synergized with chemotherapy to reduce metastatic risk. In addition, in response to chemotherapy, TAMs and inflammatory monocytes enhance CSCs’ properties of pancreatic tumors in vivo and contribute to tumor spheroid formation in vitro by directly activating the transcription factor STAT3 [41], supporting the idea that TAMs contribute to not only the metastatic structure, but also to chemo-resistant and recurrent/metastatic cancer cells. Killing tumor cells and synchronously closing the structure may greatly reduce the pro-metastatic effect of chemotherapy.

### 3.2. Micro-Metastatic Niche Induced by Chemotherapy

The pro-metastatic niche produced by chemotherapy in secondary sites is totally different from the primary foci. Chang et al. found that abundant inflammatory monocytes were recruited into metastatic lung tissue due to the upregulated chemokine CCL2 secreted by noncancer cells in which the stress-inducible gene ATF3 is upregulated [38]. MDSCs and TAMs could also be recruited into the immunosuppressive metastatic niche due to Ca3/Ca5 and colony-stimulating factor 1 (CSF-1) derived from CAFs, respectively [42,43]. These studies suggest that chemotherapeutic regimens may partly determine metastatic sites due to the heterogeneity of organ response with chemokines/cytokines secretion and myeloid cell recruitment. Thus, the chemotherapeutic regimen and metastatic site should be considered when formatting combined treatment strategies. The TME in existing metastatic sites could also be affected by chemotherapy. After stopping gemcitabine treatment, metastatic tumor cells secrete CXCL1 and CXCL2 to attract neutrophils that express growth arrest-specific 6 (Gas6) protein, a ligand of AXL receptor on metastatic tumor cells, resulting in tumor regrowth and progression. This outcome is reversed by the disruption of neutrophil infiltration or inhibition of the Gas6/AXL axis [44]. In clinical practice, it may be futile to use an immune checkpoint inhibitor for pancreatic cancer patients who stop using an ineffective first-line chemotherapy, since this strategy does not target these recruited neutrophils; furthermore, recruited neutrophil extracellular traps could even make immune checkpoint inhibitors resistant by excluding cytotoxic CD8 T cells from tumors [45]. However, a recent study demonstrated that a set of purinergic receptor P2RX1-deficient neutrophils, featured by upregulated transcription factor Nrf2 and associated PD-L1 expression, accumulate in liver metastases of untreated pancreatic cancer [46]. These neutrophils have been demonstrated to be sensitive to an anti-PD-L1 inhibitor to compromise the immunosuppressive effects on activated CD8+ T cells [46]. It is unknown whether these recruited Gas6+ neutrophils induced by chemotherapy from peripheral blood could be targeted by immune checkpoint inhibitors or not.

### 3.3. Impact of Chemotherapeutic Course and Dose

Short-term paclitaxel, but not gemcitabine, induces matrix metalloproteinase-9 expression from bone marrow-derived cells and epithelial-to-mesenchymal transition to accelerate metastasis in mice with lung carcinoma [47]. In pancreatic ductal adenocarcinoma, CAFs induced senescence-associated secretory phenotype mediators were found to be upregulated in response to short-duration of 5-FU and gemcitabine [48], leading to enhanced tumor cell viability as well as migration and invasion. A prefer pathologic complete response was observed in the arm with six cycles of neoadjuvant chemotherapy compared with three cycles for operable breast cancer; the arm, with more than five neoadjuvant chemotherapy cycles, has a poorer prognosis than those receiving 3–4 cycles in ovarian cancer [49,50]. These results imply that the duration of chemotherapy may be an important factor to determine tumor fate, and the short course may produce more factors promoting tumor progression/metastasis. However, it is difficult to define the “short course” of chemotherapy and this may also depend on tumor types.

In addition, the chemotherapeutic dose could also impact the balance of its anti-tumor and pro-tumor effects. Weekly gemcitabine with a maximum tolerated dose (500 mg/kg)—but not continuous metronomic gemcitabine (i.e., a reduced dose of weekly gemcitabine was used at 375 mg/kg)—promotes CD11b+Gr1+ myeloid-derived suppressor cells mobilization and increases angiogenesis in a pancreatic tumor model [51]. CD11b+Gr1+ myeloid cells could secret prokineticin 2 (PK2/Bv8), which contributes to tumor growth by a positive feedback loop by myeloid-derived suppressor cell differentiation into macrophages and mobilization [51]. Bv8 inhibitor and continuous metronomic gemcitabine can be added to prevent pancreatic cancer recurrence/metastasis induced by post-chemotherapy. Metronomic gemcitabine is a therapeutic concept of administering cytotoxic agents continuously at lower doses relative to maximum tolerated dose without drug-free breaks over extended periods. This administration model, compared with maximum tolerated dose, significantly increases apoptosis in cancer-associated fibroblasts, suppresses tumor growth, improves perfusion, and reduces hypoxia in human pancreatic ductal adenocarcinoma [52]. This outcome may predominantly attribute to fibroblast scavenging rather than direct tumor killing of metronomic gemcitabine. Fibroblast scavenging could increase intratumoral gemcitabine accumulation by modulating key gemcitabine metabolic enzymes [53]. These aforementioned studies are good examples to illuminate that the reshaped TME by chemotherapy with different cycles and doses could, in turn, affect their tumor killing ability.

Overall survival (OS) and recurrence free survival (PFS) were associated with the number of chemotherapy cycles received preoperatively [54]. Patients with pancreatic cancer who received ≥67% of the recommended chemotherapy cycles or ≥56% cumulative relative dose intensity had improved OS [55]. Six to eight cycles of neoadjuvant chemotherapy are recommended to treat borderline/locally advanced pancreatic cancer due to an increasing survival benefit for chemotherapy cycle up to eight cycles on univariate analysis. However, only ≥6 cycles were predictive on multivariate analysis [56]. ≥56% cumulative relative dose intensity is the recommended dose of FOLFIRINOX for patients with metastatic pancreatic cancer.

### 3.4. Strategies to Prevent Chemotherapy-Induced Tumor Recurrence/Metastasis

Tumor metastasis caused by chemotherapy is mainly mediated by immunosuppressive myeloid cells mobilized from bone marrow and peripheral blood by chemokines and cytokines. Among these myeloid cells, macrophages and neutrophils are transformable from pro-tumor to anti-tumor, suggesting that associated transformation pathways, rather than complete removal of these cells, can be operable to synergize with chemotherapy [57]. Another reason is that these cells, rather than lymphatic cells, are abundant in most pancreatic TME. Notably, patients with pancreatic cancer receiving CCR2 inhibitor (targeting tumor-associated macrophages) showed increased tumor-infiltrating CXCR2+ tumor-associated neutrophils following treatment, implying a compensatory influx of an alternative myeloid subset, resulting in a persistent immunosuppressive TME and promoting therapeutic resistance [57]. Dual targeting of CCR2 and CXCR2 improves antitumor immunity and FOLFIRINOX response in pancreatic cancer compared with either strategy alone [58]. It is presumable that the addition of anti-CTLA4/anti-PD-1/anti-PDL1 antibody to chemotherapy may be inferior to the combination of targeting molecular in myeloid cells with chemotherapy, especially for resectable pancreatic cancer after surgery in which immunosuppressive myeloid cells’ mobilization into metastatic sites by the host response may overwhelm the highly proliferative tumor cell-killing effect and cytotoxic immune cell recruitment caused by immunogenic cell death. Therapeutic efficacies of immunotherapy may be improved after the depletion of immunosuppressive myeloid cells, since it is effective to increase the intratumoral accumulation of activated T cells [59]. Nielsen et al. [60] demonstrated that lorlatinib could modulate tumor-promoting neutrophil functions and improve the response to an anti-PD-1 antibody due to more activated CD8+ T cell infiltration in pancreatic the TME.

## 4. Metabolic Substances and Exosomes Participate in the TME Remodeling

### 4.1. Metabolic Remodeling in the TME Due to Chemotherapy

To date, most studies focus on the metabolic characteristics of chemoresistant tumor cells and suggest strategies to synergize with chemotherapy by blocking compensatory metabolism [61,62,63]. For example, serine biosynthetic activity, which is important for cancer survival and proliferation, has been demonstrated to be decreased by chemotherapy [64]. However, a subset of cancer cells could adopt compensatory metabolism to resist chemotherapy. 5-FU-resistant colorectal cancer cells display a strong serine dependency achieved either by upregulating endogenous serine synthesis or increasing exogenous serine uptake [65]. Importantly, regardless of the serine feeder strategy, serine hydroxymethyltransferase-2 (SHMT2)-driven compartmentalization of one-carbon metabolism inside the mitochondria represents a specific adaptation of resistant cells to support purine biosynthesis and potentiate DNA damage response [65]. In addition, platinum-based chemotherapy could decrease the phosphoglycerate dehydrogenase (PHGDH) level to decrease serine biosynthetic activity but, simultaneously, regenerate oxidized nicotinamide adenine dinucleotide (NAD+), which helps tumor cells in sustaining poly (ADP-ribose) polymerase (PARP) activity under treatment [64]. Hence, PARP and NAD+ biosynthesis inhibition are recommended to be combined with platinum-based chemotherapy to tackle chemoresistance regardless of BRCA mutation or homologous recombination deficiency [64].

Recently, Amrutkar et al. [66] firstly provided evidence of the effects of neoadjuvant chemotherapy on pancreatic cancer metabolism with mostly lower expression of metabolic proteins at the tumor level and reduced serum lactate and high-density lipoprotein-cholesterol at the system level. By comparing differentially expressed proteins (DEPs), they filtered 3 and 46 proteins being significantly higher and lower in the neoadjuvant chemotherapy than in the treatment naive arm. These DEPs are associated with lipid metabolism, canonical glycolysis, gluconeogenesis, fatty acid (beta) oxidation, and the alpha-linolenic acid metabolic process. Additionally, eight of the differentially expressed phosphoproteins are known to be associated with metabolic processes, including purine metabolism, superoxide metabolism, and insulin-stimulated glucose transport. Reduction in metabolic tumor parameters of FDG- PET/CT after neoadjuvant chemotherapy indicates improved overall survival and recurrence-free survival [67]. They make contributions to the fields that chemotherapy indeed induces metabolic remodeling in the tumor microenvironment, while more studies are urgently required to explore these programs, including which cell types predominantly undergo metabolic remodeling, how chemotherapy reshapes metabolism, and how these remodeled metabolic products act like exosomes to net components in the TME, etc.

### 4.2. Exosome-Mediated Reshaping of the TME by Chemotherapy

Exosomes, containers of cellular debris and molecular transfer releases, could net intercellular communication among components in the TME [68]. Under doxorubicin treatment, the concentration of exosome was three-fold higher compared to healthy and untreated prostate tumor-bearing mice [69]. Consistently, Andrade et al. [70] demonstrated that human and murine melanoma cells secrete more exosomes after treatment with temozolomide and cisplatin. The roles of chemotherapy-induced exosomes depend on differences in the exosome composition and target cells. Some exosomes derived from cancer cells could be taken up by T lymphocytes to activate p38 MAPK; then, ER stress-mediated apoptosis can be induced, ultimately causing immunosuppression [71]. MiR-203 and miR-212–3p in exosomes derived from pancreatic cancer cells are involved in immune tolerance, while HSP70 promotes the migration and cytolytic activity of NK cells [71]. Exosomes induced by chemotherapy play multiple roles including acquired chemoresistance, remodeling of the TME, and even metastatic promotion.

Gemcitabine-induced exosomes could confer chemoresistance to pancreatic cancer cells by upregulation of miR-155 and miR-365 to suppress gemcitabine-metabolizing enzyme and enzyme cytidine deaminase, respectively [72]. CAFs exposed to gemcitabine significantly increase the release of exosomes containing miR-146a and Snail, which were taken up by recipient epithelial cells to promote proliferation and drug resistance. These studies show the potential for exosome inhibitors as treatment options alongside chemotherapy for overcoming chemoresistance and progression [73].

Exosomes shed by melanoma cells after temozolomide treatment promote macrophage phenotype skewing towards the M2 phenotype and favor melanoma re-growth accompanied by an increase in Arginase 1 and IL10 gene expression levels by stromal cells and an increase in genes related to DNA repair, cell survival and stemness in tumor cells [70].

In pancreatic cancer, the protein Lin28B in cancer cell-derived exosomes is transformed into these neighbor cancer cells to activate the Lin28B/let-7/HMGA2/PDGFB signaling pathway, promoting interaction with PSCs to enhance their migration to micro-metastatic sites [74]. Doxorubicin drives breast cancer cells to induce IL-33, leading to the activation of Th2 T cells with abundant IL13 release through the IL-33/ST2 signal [75]. On the other hand, it also participates in the induction of IL-13 receptors and miR-126a expressed on/in the MDSCs, which could be furtherly recruited to produce exosomes containing miR-126a by the Th2-IL-13/MDSC-IL-13R pathway. Taken together, IL-13R+miR-126a+MDSC promotes breast tumor lung metastasis through induction of IL-13+Th2 cells and tumor angiogenesis and rescues doxorubicin-induced MDSC death in an S100A8/A9-dependent manner [75]. This study is consistent with the clinical outcome that doxorubicin treatment was largely efficacious in inhibiting primary tumors, but it significantly increased the incidence and burden of pulmonary metastasis.

## 5. Malignant Cells with Cancer Stem Cell and Epithelial-to-Mesenchymal Transition Characteristics after Chemotherapy

### 5.1. Malignant Cells Reshaped by Chemotherapy

Although the quality of malignant epithelial cell decreases due to chemotherapy, the remaining cells represent chemoresistant and more malignant phenotypes, especially in CSC and partial EMT phenotypes.

Werba et al. [33] used single-cell RNA sequencing to reveal that classical and basal-such as cancer cells exhibit similar transcriptional responses to chemotherapy and do not demonstrate a shift towards a basal-like transcriptional program among treated samples. However, Hwang et al. [21] refined tumor cell types into seven malignant lineages with single nucleus RNA sequencing: neural-like progenitor; squamoid; mesenchymal; acinar-like; neuroendocrine-like; basaloid; and classical pancreatic cancer; they suggested that the neural-like progenitor (NRP) malignant program was enriched in all post-treatment groups, including organoids treated ex vivo; it was associated with worse clinical outcomes. Additionally, gemcitabine induces CXCR4 expression in pancreatic cancer cells indirectly by stimulating ROS [76], which in turn activates ERK1/2 and Akt and resultantly upregulates the nuclear factors NF-κB and HIF-1α that bind to and activate expression of the CXCR4 promoter. This reciprocal feedback loop drives tumor progression [77]. The CXCL12-CXCR4 axis signal between stromal cancer cells and cancer cells induces tumor metastasis and attenuates chemotherapy-induced apoptosis [76]. Currently, CXCR4 inhibitors such as AMD3100 and BL-8040 alone or with chemotherapy/immunotherapy are under clinical trials (NCT04177810; NCT02179970). Some completed clinical trials have proved that they were safe and well tolerated and showed signs of efficacy in some patients with advanced pancreatic cancer [78,79]. Other strategies targeting pancreatic cancer cells are difficult to develop, especially for those surviving part-MET cells and CSCs under chemotherapy.

### 5.2. The Commonality of Cancer Stem Cells and Partial Epithelial-to-Mesenchymal Transition Cells under Chemotherapy

Cancer stem cell and epithelial-to-mesenchymal transition (EMT) properties, which are associated with cancer recurrence/metastasis and chemoresistance, are key features of advanced pancreatic cancer during natural cancer progression [11]. These cell types share similar inducing conditions and signaling pathways such as hypoxia and WNT/β-catenin signaling pathway, probably due to CSCs having a plastic window to show partial-EMT characteristics [80]. They are found to be upregulated after chemotherapeutic induction by drug-resistant selection and/or reeducation by cytokines such as TGF-β [81]. Exosomes from pancreatic CSCs use agrin protein to promote Yes1-associated transcriptional regulator (YAP) activation via LDL receptor-related protein 4, resulting in tumor progression and metastasis [82]. The Hippo pathway effector YAP1 is a potent transcriptional coactivator and forms a complex with ZEB1 to activate integrin α3 transcription through the YAP1/transcriptional enhanced associate domain (TEAD) binding sites in human pancreatic cancer cells, leading to pancreatic cancer metastasis and EMT plasticity [83]. Notably, CSCs and partial-EMT cells are not necessary for pancreatic cancer tumor metastasis. Despite the reduction in partial EMT program in pancreatic cancer by the knockout of two important mesenchymal cell marker genes, metastases frequency remained unchanged, suggesting that alternative mechanisms may also support the formation of metastatic lesions [84]. However, the EMT dose induces gemcitabine chemoresistance in pancreatic cancer by mediating cancer cell proliferation with decreased expression of nucleoside transporters in tumors [84]. In spontaneous breast-to-lung metastasis models, a small proportion of tumor cells undergo EMT in lung metastases also were observed, but after chemotherapy, EMT cells significantly contribute to recurrent lung metastasis formation [85], suggesting that chemotherapy could enhance the metastatic ability of partial-EMT cell.

### 5.3. Cancer Stem Cell

Sonic hedgehog (SHH) and mammalian target of rapamycin (mTOR) signaling pathways are essential for the self-renewal of CSCs [86]. Cyclopamine, an inhibitor of the SHH pathway, significantly reduces the percentage but does not impact the viability of CD133+ stem cells, while rapamycin does not affect the content of CD133+ cells but significantly reduces the overall viability of pancreatic cancer cells [86]. These opposite outcomes suggest that the two pathways act through different mechanisms to maintain stem cells. The hypothesis that combining chemotherapy with regimens that target CSC can increase long-term survival was tested successfully in a preclinical triple therapy with gemcitabine, cyclopamine, and rapamycin [87]. However, in the phase I clinical trial, vismodegib and sirolimus failed to block signals in the SHH and PI3K/Akt/mTOR pathways and provided only limited patient benefits, suggesting that other compensatory pathways in CSCs may maintain their function, such as the JNK pathway and upregulation of transcription factor forkhead box M1 [88,89,90]. Both histone deacetylase class I inhibitor domatinostat and vitamin D receptor signaling activator 1,25-dihydroxy vitamin D3 could sensitize pancreatic cancer to chemotherapy by modulating forkhead box M1 [91,92]. Additionally, CSCs could be supported and enriched by gemcitabine re-educated mesenchymal stem cells through the C-X-C motif chemokine ligand-10 (CXCL10)-C-X-C motif chemokine receptor-3 (CXCR3) signaling pathway [93]. CSCs are an important factor in mediating tumor recurrence and metastasis under chemotherapy. However, no resultful strategies to date could tackle this problem.

### 5.4. Partial-Epithelial-to-Mesenchymal Transition Cells

Under gemcitabine treatment, a chemoresistant clone expressing EMT properties could be selected by inhibiting expression of the gemcitabine transporter ENT1 in pancreatic cancer cells via solute carrier family 39 member 4 (SLC39A4, also called ZIP4)-ZEB1 pathway [94]. In addition, gemcitabine and 5-FU could induce the EMT process through the activation of Notch-NF-κB-ZEB1, AKT-GSK3β-Snail1, and JNK-Snail2 pathway [95,96,97]. However, genetic deletion of Zeb1 in PDAC cells also leads to liver metastasis associated with cancer cell epithelial stabilization due to the enhanced collective migration of cancer cells and modulation of the immune microenvironment rather than EMT-MET process [98]. Both circulating and metastatic tumor cells identify a partial-EMT sub-population [99], implying that EMT partly participates in metastasis. Although many drugs targeting EMT transcription factors could synergize with gemcitabine, it may only act on the primary tumor but not the metastatic one; targeting the EMT may promote the mesenchymal–epithelial transition at distant metastatic sites, thereby favoring the outgrowth of metastatic tumor cells. It should be prudent to combine chemotherapy with EMT-targeting strategies, since many micro-metastases may already exist in the patient, even for patients that only have primary tumors, since chemotherapy can induce micro-metastases.

## 6. Adaptive Cancer-Associated Fibroblasts under Chemotherapy to Support Tumor Cell Survival and Metastasis

### 6.1. Cancer-Associated Fibroblasts Support Pancreatic Cancer Cells Survival

Under chemotherapeutic stress, cancer-associated fibroblasts (CAFs) can adapt themselves functionally to acquire survival potential; they can even invest this potential in pancreatic cancer. CAFs exposed to gemcitabine increase expression of senescence-associated secretory phenotype-like mediators via stress-associated MAPK signaling compared to treatment-naïve CAFs [48]. Gemcitabine-treated CAFs significantly increase the release of exosomes, which could be absorbed by pancreatic cancer to induce the chemoresistance factor, Snail [100]. In addition, the CXCL12/CXCR4/SATB-1 axis, which has been demonstrated to be upregulated by gemcitabine, mediates a reciprocal feedback loop between pancreatic cancer and CAFs to maintain their gemcitabine-resistant properties [101], and CXCL12/CXCR4 also plays roles in preventing CD8+ T cells and recruiting Tregs and macrophages’ access to tumor cells [102]. This axis supports the co-evolution of the cancer micro-ecosystem under chemotherapeutic stress, which mainly comprises cancer cells, CAFs, and immune cells. In colorectal cancer, chemotherapy-treated human CAFs express increased interleukin-17A to promote CSCs self-renewal and in vivo tumor growth [103]. Additionally, activation of CAFs by chemotherapy deposits collagen in primary pancreatic cancer by upregulating placental growth factor, a member of the vascular endothelial growth factor family, which is highly expressed in patients with poor outcomes [104]. Higher inflammatory CAFs (iCAFs) abundance in treated arm was observed. IL-1-mediated and JAK-STAT signaling in iCAFs have motivated trials of adding IL-1R blockade to FOLFIRINOX (NCT02021422). In patients treated with AG, upregulation of metallothionein genes in iCAFs was observed. Metallothionein proteins are associated with resistance to a variety of chemotherapeutics and may signal a chemoresistance mechanism. These studies validate the pro-tumor and chemoresistant roles of CAFs.

### 6.2. Cancer-Associated Fibroblasts Induce Micro-Metastatic Niche

Although the pro-tumor role of CAFs is doubtless, overall CAFs depletion could not bring clinical benefits in many studies, implying their functional heterogeneity. Nab-paclitaxel could reduce both stromal volume and cancer cells, while S-1 mainly has a cytotoxic effect against tumor cells without modulating the stroma. Kawahara et al. [105] demonstrated that impairment of the stromal defense and hyperactivation of pancreatic cancer cells by AG regimen facilitate more early liver metastasis than gemcitabine + S-1 regimen in patients with borderline resectable pancreatic ductal adenocarcinoma, suggesting the anti-tumor barrier role of stroma, which is mainly regulated by CAFs. Many classifications of CAFs have been proposed to explore their characteristics in pancreatic cancer. The most popular one defines inflammatory CAFs, myofibroblastic CAFs, and antigen-presenting CAFs. Sub-TMEs were proposed mainly according to the heterogeneity of fibroblasts, with “deserted” regions containing thin, spindle-shaped fibroblasts and “reactive” regions containing plump fibroblasts with enlarged nuclei, few acellular components, often rich in inflammatory infiltrate [14]. The matrix-rich “deserted” sub-TMEs harbored fewer activated fibroblasts and tumor-suppressive features. Yet, they were markedly chemoprotective and enriched upon chemotherapy [14]. After primary breast cancer was treated with doxorubicin, but not cisplatin, CAFs in the lung secrete complement factors such as Ca3 and Ca5 to recruit myeloid-derived suppressive cells to modulate the immunosuppressive metastatic niche (the soil) for circulating tumor cells (the seedlings) [43]. Likewise, another study demonstrated that in liver metastases of colorectal cancer, chemotherapy decreases the abundance of ECM-remodeling CAFs, which express ECM proteins (such as ECM collagens and fibronectin) and ECM proteases [106]. These findings suggest that CAFs in either primary or secondary TMEs may be reshaped by chemotherapy to hinder treatment efficacy and promote metastatic relapse.

Pancreatic cancer recurrence/metastasis under chemotherapy partly contributes to adaptive CAFs, making them sound and attractive targets, and the goal has been shifting from “CAF/stroma ablation” to “CAF/stroma normalization”. This strategy asks for reasonable classification to explore CAFs’ characteristics and exploration of targetable biomarkers. To date, many strategies targeting CAFs themselves and surface receptors/soluble proteins responsible for their cross-talks with cancer or immune cells are under validation in preclinical and clinical trials, and most clinical trials are just in phase I/II, suggesting the difficulty to target CAFs. These strategies are fully illuminated by Liu et al. [102].

## 7. Targeted Therapy to Synergize with Chemotherapy

Currently, numerous clinical trials are conducted to explore chemotherapy to synergize with targeted therapy, where associated molecules are associated with chemoresistance and poor survival in pancreatic cancer [107,108,109]. Sun et al. [110] have explicitly illuminated many small molecules in pancreatic cancer, including focal adhesion kinase (FAK), which is a ubiquitously expressed nonreceptor tyrosine kinase expressing in both stromal and epithelial cells. It has been shown to promote fibrotic stromal reaction including matrix remodeling and induce tissue stiffness. FAK-dependent fibrotic stromal reaction and collagen IV deposition are responses to chemotherapy in pancreatic cancer. Hence, FAK inhibitor is predicted to synergize with chemotherapy by promoting cancer cell apoptosis and altering stromal reaction [111,112]. FAK inhibitor increases the proportion of cells in the S-G2-M cell cycle induced by AG in pancreatic cancer [111] and increases 5-FU-induced caspase-3 activity in a p53-dependent manner in gastric cancer [113]. Both indicate an enhancement of chemotherapeutic efficiency. FAK inhibition could also diminish the immunosuppressive MDSCs, TAMs, and Treg, rendering previously unresponsive pancreatic cancer responsive to chemotherapy and immunotherapy [114]. Cyclin-dependent kinase 1 (CDK1) also plays key roles in cell cycle progression, resistance to the induction of apoptosis, and immune evasion [115,116,117], implying that CDK1 inhibitor may synergize with chemotherapy similarly to the FAK inhibitor.

Pancreatic cancer cell lines highly resistant to chemotherapy are also resistant to all the tested inhibitors of EGFR, AKT, and PI3K due to compensatory pathways associated with anti-apoptosis [118,119], which may partly explain the failure of combining chemotherapy and targeted therapy. However, Peng et al. [119] found that the gemcitabine-resistant (GR) cell line was more sensitive to everolimus than the gemcitabine-sensitive (GS) cell line, since high levels of Thr389- and Thr371-phosphorylated p70S6 protein (a mTOR substrate) were found in GR cells treated with gemcitabine, supporting the idea that the inhibition of p70S6 protein phosphorylation by everolimus could synergize with gemcitabine.

Chemotherapy induces mammary epithelial cells to produce monocyte/macrophage recruitment factors, including colony stimulating factor 1 (CSF1) and interleukin-34, which, together, enhance CSF1 receptor (CSF1R)-dependent macrophage infiltration [120]. Macrophages further produce IL-10 to block CD8+ T cell-dependent responses to chemotherapy by suppressing IL-12 expression in intratumoral dendritic cells. Hence, blockade of colony-stimulating factor-1 (CSF-1) limits macrophage infiltration and improves response of mammary carcinomas to chemotherapy [120]. CAFs are also known to recruit and polarize macrophages in a CSF1–CSF1R dependent manner, but low levels of the CSF1–CSF1R interaction did not change with treatment in pancreatic cancer, while the interaction between CXCL12 (on iCAFs but not myCAFs) and CXCR4 (on both TAM subpopulations) significantly weakened with treatment [33,121]. Notably, it appears that the change in the volume of the CXCL12–CXCR4 interaction is driven by a decrease in CXCR4 in TAMs with treatment, which implies that chemotherapy acts partly as CXCR4 inhibitors to weaken macrophages infiltration.

Only some of the macrophages express either CSF1 or CXCR4, meaning that the remaining macrophages could also impair the response of chemotherapy with other pathways. Veracious macrophage subtypes, with their properties remodeling during chemotherapy, are the next topic of exploration. In addition, chemotherapy reduces inhibitory checkpoint molecules in pancreatic cancer including CD80/CD86-CTLA4, LGALS9-HAVCR2, DPCD1LG2-PDCD1, and TIGIT-PVR [33,121]. These outcomes could partly explain chemotherapy with checkpoint inhibitory drugs such as PD-1/PD-L1 inhibitors showing modest benefit in clinical trials, and TIGIT inhibitors may be more proper as alternative immunotherapy for pancreatic cancer. TIGIT is the highest and most broadly expressed inhibitory checkpoint molecule and decreases in CD8+ T cells in treated samples, while PDCD1 is expressed at low levels and in a minority of CD8+ T cells.

Some selected clinical trials of combination strategy of immunotherapy/targeted therapy with chemotherapy are listed in Table 1.

## 8. Conclusions

The reshaped tumor microenvironment is an outcome impacted by the baseline tumor microenvironment (dynamic with tumor progression) and chemotherapy regimens (including drug, course, and dosage). Generally, neoadjuvant chemotherapy could invert natural tumor progression, leading to decreased immunosuppressive myeloid cells and increased anti-tumor lymphatic cells, which are mainly attributed to chemotherapy-induced immunogenic cell death. Alternatively, more metastatic structures (i.e., microanatomical tumor microenvironment of metastasis) in primary tumors could be induced by chemotherapy to leak circling tumor cells into peripheral blood and recruit immunosuppressive myeloid cells. Meanwhile, the host response caused by chemotherapy provides micro-metastatic niche-homing for circling tumor cells. Responsive biomarkers in peripheral blood, such as circling tumor cells, immune cells, and cytokines, are suggested to be monitored dynamically during chemotherapeutic treatment. Novel/upregulated molecules, such as TIGIT, CXCR4, and CSF-1, due to treatment could be targeted to synergize with neoadjuvant chemotherapy or prevent adverse effects. Additionally, some tumor clones with cancer stem cell and epithelial-to-mesenchymal transition properties and subgroups of cancer-associated fibroblasts selected and/or educated by chemotherapy could adapt to the reshaped tumor microenvironment; this is another reason for tumor recurrence and metastasis. Progressed pancreatic cancer is attributed to the coordinated evolution of malignant and non-malignant cells in the sub-tumor microenvironment (Table 2), which is mainly composed of pancreatic cancer cells, cancer-associated fibroblasts, and immune cells. The focus has been shifting from how the overall TME impacts chemotherapy to how novel/upregulated molecules in the reshaped TME at different stages are targeted to synergize with chemotherapy or prevent chemotherapy-induced recurrence/metastasis. The cycle and dosage of chemotherapy should also be considered when formatting combination therapy.

## Figures and Tables

**Figure 1 cancers-15-02448-f001:**
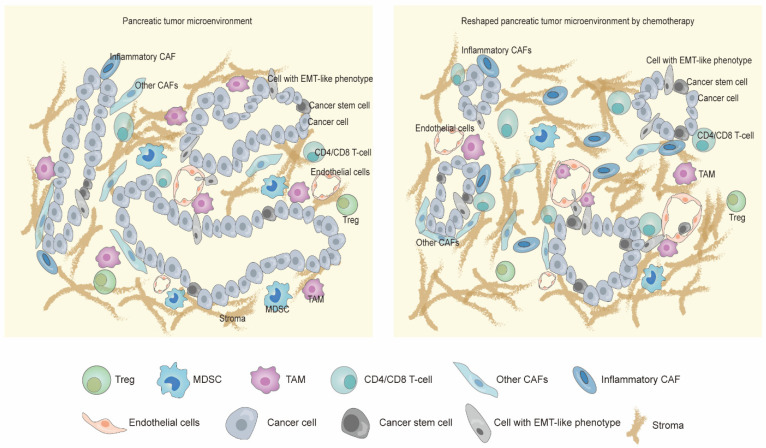
The pancreatic cancer microenvironment is reshaped by chemotherapy.

**Figure 2 cancers-15-02448-f002:**
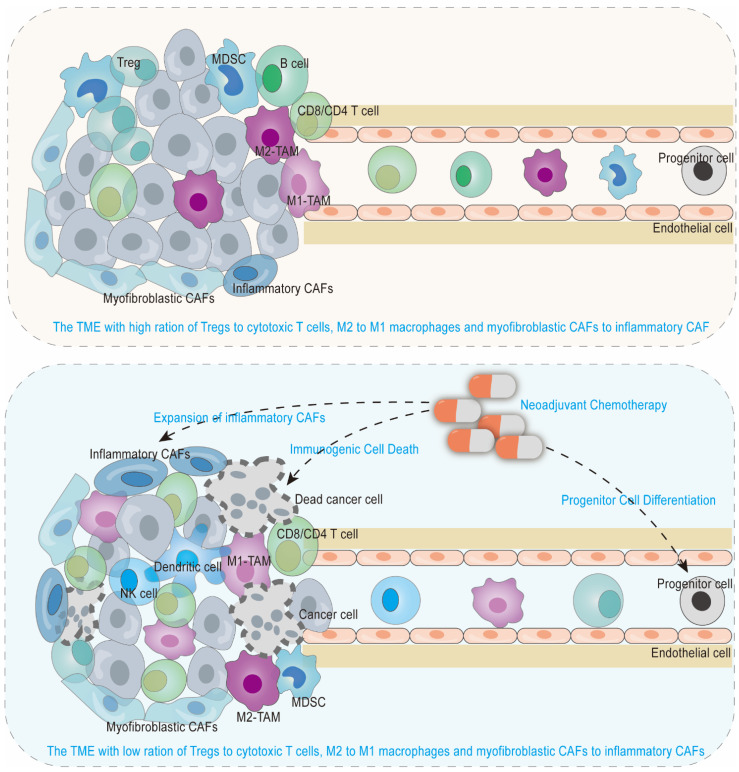
Neoadjuvant chemotherapy induces an anti-tumorigenic microenvironment via immunogenic cell death and progenitor cell differentiation, as well as expansion of inflammatory CAFs.

**Figure 3 cancers-15-02448-f003:**
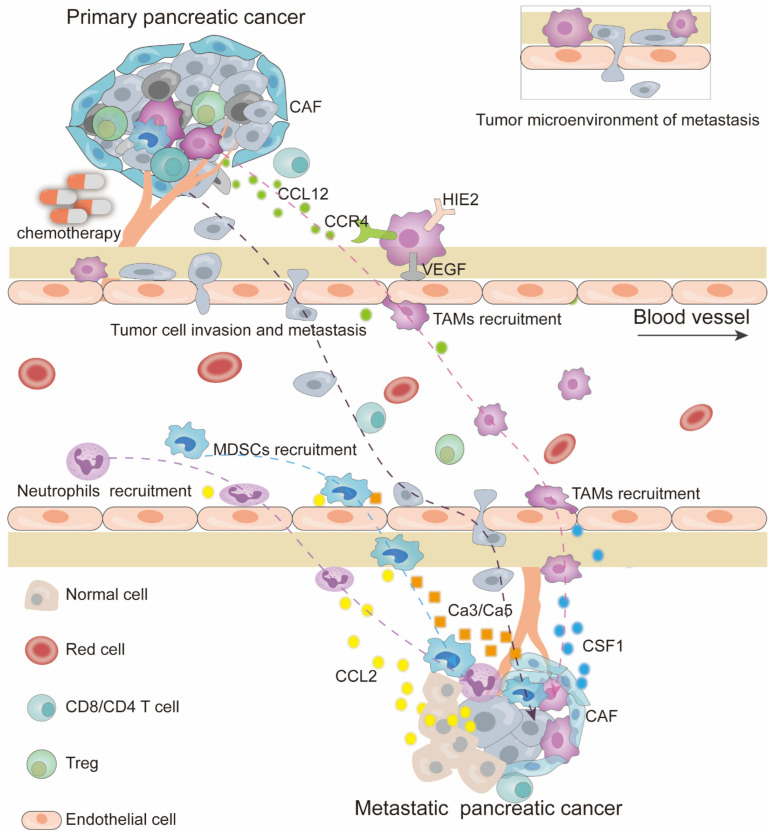
Chemotherapy promotes disease progression and induces pancreatic cancer metastasis.

**Table 1 cancers-15-02448-t001:** Selected clinical trials of combination strategy of immunotherapy/targeted therapy with chemotherapy.

Combined Treatment Strategy	Target	Target Inhibitor	Chemotherapy	Other Regimens	Phase	(Estimated) Number Enrolled	NCT Number PMID Number
Immunotherapy	PD-1	Camrelizumab	NabPaclitaxel/Gemcitabine	NA	Phase 2	117	NCT04498689
Immunotherapy	PD-1	Camrelizumab	NabPaclitaxel/Gemcitabine	NA	Phase 3	401	NCT04674956
Immunotherapy	PD-1	Pembrolizumab	Gemcitabine	Defactinib	Phase 1	43	NCT02546531
Immunotherapy	PD-L1	Durvalumab	Nab-Paclitaxel/Gemcitabine	Oleclumab	Phase 2	30	NCT04940286
Immunotherapy	CTLA-4	Ipilimumab	Gemcitabine	NA	Phase 1b	21	NCT01473940
Immunotherapy	CTLA-4	Ipilimumab	FOLFIRINOX	Vaccine	Phase 2	83	NCT01896869
Immunotherapy	CTLA-4	Zalifrelimab	Nab-Paclitaxel/Gemcitabine	NLM-001	Phase 1/2	24	NCT04827953
Immunotherapy	LAG3	IMP321	Gemcitabine	NA	Phase 1	18	NCT00732082
Targeted therapy	CXCR4	Motixafortide	Irinotecan/Fluorouracil /Folinic Acid	pembrolizumab	phase 2a	80	NCT02826486
Targeted therapy	CXCR4	Motixafortide	Onivyde/Leucovorin/5FU	pembrolizumab	Phase 2a	137	NCT02826486
Targeted therapy	CCR2	PF-04136309	FOLFIRINOX	NA	phase 1b	44	NCT01413022.
Targeted therapy	CCR2	PF-04136309	Nab-Paclitaxel/Gemcitabine	NA	phase 1b		NCT02732938
Targeted therapy	PARP	Veliparib	5fluorouracil/oxaliplatin	NA	Phase 1/2	64	NCT01489865
Targeted therapy	EGFR	Erlotinib	Gemcitabine	NA	Phase 3	569	PMID:17452677
Targeted therapy	EGFR	Nimotuzumab	Gemcitabine	NA	Phase 2b	192	PMID:28961832
Targeted therapy	EGFR	Cetuximab	Gemcitabine	NA	Phase 2	41	PMID:15226328
Targeted therapy	EGFR	Matuzumab	Gemcitabine	NA	Phase 1	17	PMID:16622465
Targeted therapy	VEGF	Bevacizumab	Gemcitabine/capecitabine	erlotinib	Phase 1/2	44	PMID:24613126
Targeted therapy	VEGF	Bevacizumab	Gemcitabine	NA	Phase 2	52	PMID:16258101

**Table 2 cancers-15-02448-t002:** Regional heterogeneity of tumor microenvironment in pancreatic cancer.

Characteristics of Various Sub-Microenvironment		
Sub-Microenvironment	Intermediated/Reactive Submicroenvironment	Deserted Sub-Microenvironment
CAFs characteristics	CAFs dedifferentiated	CAFs activated and pro-inflammatory
Stroma	ECM-rich stroma	ECM-rare stroma
Overall immune features	Immune-cold	Immune-hot
Immune cell distribution	Small/Scattered immune clusters	Uninterrupted clusters with all immune cells
Main immune cell type	CD20 B cells	CD3/CD8 T cells; CD68 macrophages
Tumor cells feature	Poorly differentiated tumor cells	Well-differentiated tumor cells
Response to chemotherapy	Sensitive to chemotherapy	Resistant to chemotherapy
Cellar stress response	Rare	Heat shock; Hypoxia; Metabolic stress

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
