# Peer review of "Reshaping the Pancreatic Cancer Microenvironment at Different Stages with Chemotherapy"

_cancers, 2023, doi:10.3390/cancers15092448_

Round 1

Reviewer 1 Report

Manuscript ID: cancers-2305602

Type of manuscript: Review

Title: Reshaping the pancreatic cancer microenvironment at different stages with chemotherapy

Summary: In this article, the authors reviewed published literature regarding TME changes upon chemotherapy and reshaping TME. Reshaped pancreatic cancer tumor microenvironments due to chemotherapy were reflected mainly in immune cells, pancreatic cancer cells, and cancer-associated fibroblasts cells, quantitatively, functionally and spatially. The authors conclude that an in-depth understanding of how chemotherapy reshapes the tumor microenvironment may lead to new therapeutic strategies proposed to improve its efficacy and block its negative tumor-promoting effects, resulting in the expansion of potential resectable pancreatic cancer patients, elimination of micro-metastases, and prolonged survival.

Comments:

1.     The authors need to include metabolic remodeling in the tumor microenvironment due to chemotherapy.

2.     The authors need to include exosome-mediated reshaping of TME in chemoresistance.

3.     What are circling cancer cells?  Are these same as circulating tumor cells? Nomenclature for consistency is important.

4.     Line 72-74: the authors state “Additionally, sub-tumor microenvironments (sub-TMEs) have been proposed to validate their roles in propelling malignant cell progression (Table 1).” Table 1 shows “Selected clinical trials of combination strategy of immunotherapy/targeted therapy with chemotherapy”. The connection between line 72-74 and table 1 not clear. Please discuss the table 1 in the proper section and appropriately.

5.     Lines 508-509”: During chemotherapeutic treatment, novel/upregulated molecules, like PD-1/PDL1, CXCR4 and CSF-1, accompanied by chemotherapy could be targeted to synergize with neoadjuvant chemotherapy or prevent adverse effects (Table 2). Table 2 titled as “Regional heterogeneity of tumor microenvironment in pancreatic cancer”. Is table 2 cited correctly here?

6.     The review article will benefit from review by a scientific english writer. Frequently, the sentence construction seems complex, and the underlying meaning of the sentence is not clear.

Author Response

Thank you for your positive comments and valuable suggestions to improve the quality of our manuscript. According to your valuable comments and suggestions, we have made extensive modifications to our manuscript to make our arguments convincing. 

Reviewer 2 Report

The introduction is expertly written, although the short description of the clinical therapy regimens used could be explained more broadly so that non-clinicians can better understand the basis of these therapies. 

The description of the tumor immune microenvironment also is expertly written and provides a lot of information between just a few lines. Also here, I could imagine that being slightly more elaborate about which immune cells are involved at which states of tumor progression would be an excellent asset for the review (and the reader). This information provides the basis for the entire rest of the article and therefore, should be precise and detailed and fully cover the topic. A lot of terms are used here, such as "immune hot", that may not be familiar to everone, and need more explanation and background. 

Figure 1: this picture shows a steady state scenario, how is this RESHAPED by chemotherapy? That would require 2 images, one before, and another during or after therapy. Or, in some way, it must be indicated that is changed. 

Figure 2:  This image takes a while to understand, and it may be clearer if the cells in question were simply labelled IN the figue itself. Otherwise, its a bit difficult to make out which is which from the legend below.  

The next paragraph is fine, introducing all the most important and relevant immune cells that feature in immune responsive versus immunosuppressive tumors, such CD4+ and CD8+ T cells, NK cells, Tregs, M1 versus M2 macrophages or myeloid cells, MDSCs etc. It just needs to be somewhat cleaned up because the English language use is a bit challenging here. 

Its also nice to see that the authors have not forgotten the role of cancer-associated fibroblasts in the question of how immunoresistant tumors may emerge. But they could also be mentioned/shown somewhere on Fig. 2? 

Generally, there is a lot of focus on the immune cells in pancreatic cancer, and that's fine if that is what the authors want to focus on. However, in my opinion, the suppressive role of the TME and ECM generated by cancer-associated fibroblasts, and the problem of desmoplastic TME, and fibrosis etc. This may be of higher relevance than we think, and could therefore attain a bit more attention here. Currently, it is dealt with in the last paragraph prior to conclusions. 

Figure 3 is very complex, but it tries to recapitulate a very complex piece of biology. It takes quite some time to understand all the little issues conveyed in Fig. 3 but I dont have any suggestion on how to improve it. The good thing is that this figure goes along with the text in the following paragraph which quite nicely explains the changes in TME induced by chemotherapy. 

It is also nice that the authors bring up the connection to clinical issues, such as the dosing of the drugs and their changing impact on TME and metastasis (paragraph 3.3). This is not reviewed anywhere else, to my knowledge, and could therefore even get more attention in the article. 

Paragraph deals with the general subject of "tumor cell plasticity", which is common in pretty much all advanced epithelial cancers, and - in my opinion - doesn't really add much NOVEL to the big picture. Its fine to be here as it is part of the big picture. And as usual, the features of increasing EMT nd increasing tumor stem cell characteristics, or stemness, are a fluid concept. They simply reflect that there is increasing plasticity in how the tumor cells adapt to a challenging environment. But as the authors not themselves - this plasticity is NOT a prerequisite for metastasis to occur. It should therefore be handled carefully. Especially cancer stem cells are poorly understood in pancreatic cancer and I think some of the discussion here is not far from speculation. 

but altogether, I think the authors have handled these partially controversial discussions remarkably well. 

But whats wrong with Figure 4? It says that 3 different states of the tumor microenvironment are shown - but we see only 2 of them. Maybe this figure isnt necessary. 

small issues:

there are some repeated grammatical errors throughout the manuscript, please use a program such as "Grammarly" or a simple thesaurus to fix these somewhat annoying errors. One of the is the use of the word "impact": it always has to come in combination with the word "one": impact ON something... or HAVE an impact ON... 

Another example is the term tumor microenvironment (TME). It should always be referred to by the term THE tumor microenvironment. But it never is in this paper 

line 20: "Unsorted pancreatic cancer patients" - this needs to mean non-stratified or non-classified (use scientific terms) 

line 22: "increasing NUMBER of studies" 

line 28: "invade peripheral blood," this must mean invading the lymph or blood vasculature - nobody invades the peripheral blood 

line 49: describe what R0 incidence means 

line 31:"neglectful" cannot be used here, it describes what persons do 

line 84/85: "genetic mutation and microbial could also attribute to TME heterogeneity"... microbial WHAT? this remains unclear 

Author Response

(The authors gave the same response as above.)

Reviewer 3 Report

Query#1

The introduction was found to be too confused, for instance, in the line 46-51 the authors stated that the first line regimen for borderline and locally advanced patients is FOLFIRINOX and AG (gemcitabine and nab-paclitaxel), at the same time is also stated that the first line regimen for metastatic and advanced patients is also FOLFIRINOX and AG. Please clarify this point.

In addition, in the introduction, considering the importance of gemcitabine in the chemotherapeutic regimens. I suggest to the authors to mention the role of human equilibrative nucleoside transporter 1 (hENT-1) as a biomarker for gemcitabine efficacy in PDAC.

At this purpose I suggest to the authors to cite this update review:

-Randazzo, O., Papini, F., Mantini, G., Gregori, A., Parrino, B., Liu, D. S. K., Cascioferro, S., Carbone, D., Peters, G. J., Frampton, A. E., Garajova, I., & Giovannetti, E. (2020). "Open Sesame?": Biomarker Status of the Human Equilibrative Nucleoside Transporter-1 and Molecular Mechanisms Influencing its Expression and Activity in the Uptake and Cytotoxicity of Gemcitabine in Pancreatic Cancer. Cancers12(11), 3206. https://doi.org/10.3390/cancers12113206

Query#2

Page 2: please clarify the sentence reported in lines 77-81 and explain how the different clinical stages could influence the reshaping of TME and the efficacy of standard chemotherapeutic drugs.

Query#3

In the introduction the authors reported that the reshaping of tumor microenvironment and the consequent upregulation of different targets may provide novel treatments. For instance the authors reported the role of immune checkpoint inhibitors (ICIs) as adjuvant treatments in PDAC. However, in the last years different small molecules kinase inhibitors (SMKIs) are currently tested alone or in combination with standard chemotherapeutic drugs for the treatment of pancreatic cancer, therefore I suggest to the authors to mention the importance of SMKIs aside from ICIs.

At this purpose, I suggest to the authors to cite the following updated literatures:

1)     Sun, J., Russell, C. C., Scarlett, C. J., McCluskey, A. (2020). Small molecule inhibitors in pancreatic cancer. RSC medicinal chemistry, 11(2), 164–183. https://doi.org/10.1039/c9md00447eI

2)     Jiang, H., Hegde, S., Knolhoff, B. L., Zhu, Y., Herndon, J. M., Meyer, M. A., Nywening, T. M., Hawkins, W. G., Shapiro, I. M., Weaver, D. T., Pachter, J. A., Wang-Gillam, A., DeNardo, D. G. (2016). Targeting focal adhesion kinase renders pancreatic cancers responsive to checkpoint immunotherapy. Nature medicine, 22(8), 851–860. https://doi.org/10.1038/nm.4123

3)     Pecoraro, C., Parrino, B., Cascioferro, S., Puerta, A., Avan, A., Peters, G. J., Diana, P., Giovannetti, E., Carbone, D. (2021). A New Oxadiazole-Based Topsentin Derivative Modulates Cyclin-Dependent Kinase 1 Expression and Exerts Cytotoxic Effects on Pancreatic Cancer Cells. Molecules (Basel, Switzerland), 27(1), 19. https://doi.org/10.3390/molecules27010019.

#General comment

1)     Paragraph: “2.3. Reshaped tumor immune cells in quantity, space and function”, line 159 please correct “cell” in “cells”.

Overall, I appreciated this review in which the authors have clearly highlighted the role of tumor microenvironment in the effectiveness of chemotherapy in pancreatic cancer. Several recent works also showed that the tumor microenvironment and hypoxia contribute to high rate of pancreatic cancer resistance. I therefore fully recommend this publication after the revisions I suggested.

Author Response

(The authors gave the same response as above.)

Round 2

Reviewer 1 Report

The authors have addressed our concerns in the cover letter.